# Flexible and Effective Preparation of Magnetic Nanoclusters via One-Step Flow Synthesis

**DOI:** 10.3390/nano12030350

**Published:** 2022-01-22

**Authors:** Lin Zhou, Lu Ye, Yangcheng Lu

**Affiliations:** State Key Laboratory of Chemical Engineering, Department of Chemical Engineering, Tsinghua University, Beijing 100084, China; zhoul20@mails.tsinghua.edu.cn (L.Z.); jennsie16@163.com (L.Y.)

**Keywords:** magnetic nanoclusters, micromixing, one-step synthesis, flow synthesis, oleic acid modification

## Abstract

Fe_3_O_4_ nanoclusters have attractive applications in various areas, due to their outstanding superparamagnetism. In this work, we realized a one-step flow synthesis of Fe_3_O_4_ nanoclusters, within minutes, through the sequential and quantitative introduction of reactants and modifier in a microflow system. The enhanced micromixing performance enabled a prompt and uniform supply of the modifier oleic acid (OA) for both nanoparticle modification and nanocluster stabilization to avoid uncontrolled modified nanoparticles aggregation. The size of the nanoclusters could be flexibly tailored in the range of 50–100 nm by adjusting the amount of OA, the pH, and the temperature. This rapid method proved the possibility of large-scale and stable production of magnetic nanoclusters and provided convenience for their applications in broad fields.

## 1. Introduction

Magnetic nanoparticles have attracted extensive attention owing to their outstanding properties [1,2]. Among other metal nanoparticles, Fe_3_O_4_ magnetic nanoparticle has been a specially popular category that is widely applied in various fields. These fields include targeted drug delivery [3], magnetic hyperthermia [4,5], photothermal therapy [6], contrast agent in magnetic resonance imaging (MRI) [7,8], biosensor [9], magnetic field-assisted separation [10,11], catalyst [12,13,14], and so on. Most of the applications utilized the magnetism of Fe_3_O_4_ nanoparticles, putting forward a high demand for their magnetic response.

In order to utilize the advantage of superparamagnetism instead of ferromagnetism of the bulk magnet, Fe_3_O_4_ with a size smaller than 15 nm is usually required in practice [15]. However, the magnetism of individual nanoparticles is usually too weak to make an effective magnetic response. Moreover, individual nanoparticles tend to aggregate owing to the strong surface–surface interaction and the large surface-to-volume ratio, which has a negative impact on long-term applications [16,17,18]. To resolve these problems, the controllable assembly of nanoparticles into nanoclusters, which are larger clusters containing a finite number of nanoparticles, might provide an appropriate solution [19].

Various methods have been proposed to prepare Fe_3_O_4_ nanoclusters with controllable size and shape. They could be generally summarized into two categories, namely in situ synthetic method and post-assembly of ligand-grafted nanoparticles. The representatives of the former category include solvothermal synthesis and thermal decomposition. For example, Xu [20] employed various biopolymers as structure-directing ligands to synthesize nanocrystal clusters through solvothermal synthesis and revealed the function of the self-sacrificing template of the biopolymeric ligand. Moreover, the prepared nanoclusters were applied as drug delivery vehicles to encapsulate drugs for chemotherapy for cancers. Aleksey [21] synthesized magnetite nanoclusters that could be used as contrast agents for MRI by the thermal decomposition method in one step and investigated the effect of organic acid as surfactant on the size and shape of the products. Besides the above substrates, sodium citrate [22] and sodium acetate [23] were frequently utilized as stabilizers in subsequent works. Although magnetite nanoclusters could be synthesized within one step with tunable size and morphology, the process is still challenged by some problems, such as expensive reagents, low production efficiency, poor atomic economy, and so on.

As for the post-assembly, in general, Fe_3_O_4_ nanoclusters were synthesized with preformed Fe_3_O_4_ nanoparticles, and an external trigger was employed to the system to modify the interparticle forces and initiate assembly [24]. Some processes applied encapsulation or grafting to realize assembly. Kim [25] encapsulated iron oxide nanoparticles with amphiphilic block-copolymer and obtained magnetomicelles through the crosslinking of polyacrylic acid (PAA) shells. Larken [26] obtained uniform clusters containing approximately 20 particles each, where block copolypeptide poly(EG_2_-lys)_100_-*b*-poly(asp)_30_ was bounded to the surface of the nanoparticles through electrostatic interactions to form a micelle shell that is capable of controlling the size of the clusters through altering the composition of the block copolypeptide. Zhuang [27] obtained polar-solvent-dispersible Fe_3_O_4_ nanoclusters by using solvophobic interactions. Fu [28] applied a ligand-stripping method to remove the original capping ligand with diols and induced the creation of secondary nanoclusters with different sizes. Some works have also introduced operation modules to synthesize Fe_3_O_4_ nanoclusters. For example, Chang [29] used the membrane emulsification and solvent pervaporation method to synthesize magnetic nanoclusters with sizes between 100 and 300 nm continuously and coated the clusters with silica for further functionalization. In summary, colloidal interactions were used in post-assembly, including Van der Waals force, electrostatic, ligand interactions, and so on, which could be manipulated under a mild environment, but involved with complex steps and additional solvents or functional ligands, nevertheless.

Of all the methods, controlling the surface properties of primary particles and the interaction between particles is a common key point. The agglomeration caused by the interaction between nanoparticles is a rapid and sensitive process that requires efficient control methods. In our previous work, water-dispersible Fe_3_O_4_ nanoclusters were obtained via a micromixer, which could realize rapid solvent exchange and accurate size control [30]. This method benefited from the outstanding mixing performance of the micromixer, which could achieve the even and sufficient allocation of surfactant in a short time. This inspired us to start from the nano-precipitation process and realize end-to-end continuous preparation of Fe_3_O_4_ nanoclusters by precisely tailoring the grafting behavior of oleic acid (OA) onto the surface of the nanoparticles during the modification and aqueous self-assembly process online [31,32,33,34,35,36,37,38,39,40]. This method opened up a brand-new opportunity to realize the preparation of bilayer OA-coated Fe_3_O_4_ nanoclusters with stable dispersion and adjustable size in an aqueous phase and facilitate their subsequent application to a great extent.

## 2. Experimental

### 2.1. Materials

Both anhydrous iron(III) chloride (FeCl_3_, 98%) and iron(II) chloride tetrahydrate (FeCl_2_·4H_2_O, 99%+) were obtained from Acros (Geel, Belgium). Ammonium hydroxide (NH_3_·H_2_O, 28 wt%) was obtained from Peking Reagent (Beijing, China). The surface ligand oleic acid (C_18_H_34_O_2_, 90%+) was purchased from Alfa Aesar (Lancashire, UK). All the above chemicals were used without further purification. Water used throughout the experiment was prepared by an ultrapure water system (Center 120FV-S, The lab, Shanghai, China).

### 2.2. Characterization

The crystal phase of the cluster was analyzed by using an X-ray powder diffractometer (XRD, D8-Advance, Bruker, Berlin, Germany). The morphology of the cluster was observed with a transmission electron microscopy (TEM, JEM2010, JEOL, Tokyo, Japan). Before TEM measurement, the dispersion sample was diluted with water to a solid content of 0.2 mg/mL, cast onto the carbon films, and dried at room temperature. The size distribution of the clusters was measured by using dynamic light scattering (DLS, Malvern Panalytical, Malvern, UK). Herein, the polydispersity index (PDI) was applied to represent the monodispersity of the cluster, which was calculated by using Equation (1)

(1)
$$\mathrm{PDI}=\frac{\sigma^{2}}{{Z_{D}}^{2}}$$

where *Z*_D_ is the intensity weighted mean hydrodynamic size of the collection of particles measured by DLS, and *σ* is the standard deviation. To determine the amount of organic grafting on the surface, thermogravimetric analysis was carried out with a STA 409 PC apparatus (TGA, NETZSCH, Selb, Germany) at a heating rate of 10 °C·min^−1^, from room temperature to 600 °C, under a nitrogen atmosphere. The magnetic properties of the cluster were measured by a superconducting quantum interference device (VSM, SQUID, Quantum Design, San Diego, CA, USA). Fourier-transform infrared spectroscopy (FTIR, Nexus 670, Nicolet, Madison, WI, USA) was used to characterize the surface of the cluster. The samples were ground with KBr and pressed to the tablet before measurement. The contact-angle-measuring instrument (CA, OCAH200, Dataphysics, Filderstadt, Germany) was used to characterize the hydrophilicity of the cluster. The surface composition of the cluster was investigated by X-ray photoelectron spectroscopy (XPS, PHI5300, Ulvac-Phi, Chigasaki, Japan), and the XPS split-peak fitting process was carried out by using the method used in the literature [41].

### 2.3. Continuous Synthesis of Bilayer OA-Coated Fe_3_O_4_ Nanoclusters

A continuous co-precipitation, in situ modification, and self-assembly method in an aqueous solution were exploited for the preparation of finite-sized Fe_3_O_4_ nanoclusters. The schematic of the experimental setup is shown in Figure 1. The core of the setup is the membrane dispersion microreactor reported in our previous work [42], where the geometric size of the microchannel is 20 mm × 20 mm × 4 mm (length × width × height). We used a 316 L stainless-steel membrane with an average pore diameter of 5 μm as the dispersion medium. In a typical synthesis, a mixing aqueous solution containing Fe (II) (0.2 M) and Fe (III) (0.4 M) was used as the continuous fluid, and 7.3 M ammonia hydroxide was used as the dispersed fluid. Both of them were delivered at the flow rate of 10 mL/min to mix in the first membrane dispersion microreactor to generate precipitates immediately. The aqueous solution containing bare Fe_3_O_4_ nanoparticles flowed through a delayed tube with an 18.84 s residence time (residence time 1) and entered the second membrane dispersion microreactor as a continuous fluid, where the dispersed fluid of ammonia solution containing 0.1 M OA was introduced at a flow rate of 10 mL/min, to mix and trigger the modification and self-assembly process. After the slurry was aged for 12.56 s (residence time 2), a black-brown stable water-based magnetic fluid was obtained. The whole process was conducted at 60 °C, within a water bath.

## 3. Results and Discussion

### 3.1. Feasibility of Continuous Preparation of Bilayer OA-coated Fe_3_O_4_ Nanoclusters

Under the typical synthetic conditions described above, the products were characterized by using XRD, XPS, TEM, FTIR, TGA, and VSM to determine their composition, structure, morphology, and surface properties. Figure 2 shows the XRD patterns of the samples. The diffraction peaks at (220), (311), (400), (422), (511), (440), (620), and (533) are the characteristic peaks of the Fe_3_O_4_ crystal with a cubic spinel structure (JCPDS85-1436). It is known that Fe_3_O_4_ can be oxidized to γ-Fe_2_O_3_, and γ-Fe_2_O_3_ can be further transformed into α-Fe_2_O_3_ at a higher temperature. The diffraction peaks of γ-Fe_2_O_3_ and α-Fe_2_O_3_ are at (110), (113), (210), and (213) [43]. In this work, the Fe 2p XPS spectrum (Appendix A) could be successfully fit to three main peaks and two satellite peaks, and no peak could be attributed to impurities, such as γ- or α- Fe_2_O_3_. The lowest binding energy peak at 710.6 eV is attributed to Fe^2+^, with a corresponding satellite peak at 717.5 eV. A binding energy of the Fe^3+^ octahedral species was found at 711.6 eV, while that of tetrahedral species was 714 eV. These values are comparable to those in References [44,45], and, Fe^2+^/Fe^3+^ = 0.54, slightly greater than the 0.5 expected from the stoichiometry of Fe_3_O_4_. The above results showed that, although OA was introduced in a short time, it did not affect the formation of Fe_3_O_4_ nanoparticles with a complete crystal structure and even prevented further oxidation. Additionally, according to Scherrer’s formula, the mean size of crystal grains was 8.84 nm, which met the requirement of diameters lower than 15 nm for superparamagnetism.

From the TEM images (Figure 3a,b), it could be seen that the samples on the carbon film were quasi-spherical aggregates composed of dozens of Fe_3_O_4_ nanoparticles, and the lattice fringes had spacings of 0.25 and 0.50 nm (Figure 3c) corresponding to the (311) and (111) crystal planes of Fe_3_O_4_, respectively. The size measurement using DLS showed that the average hydraulic diameter of the clusters was 47.9 nm (PDI = 0.165) (Figure 3d), which was approximately equal to the size of the double OA layers (about 3.3 nm [46]) and the magnetite core (about 42 nm in the results of TEM). This initially implied the success of the in situ modification of OA.

Figure 4a showed the FTIR spectrum of the samples. The bare Fe_3_O_4_ nanoparticles had a strong absorption peak near 590 cm^−1^, which corresponded to the stretching vibration of the Fe–O bond. The bands at 2923 and 2853 cm^−1^ were attributed to the symmetric and asymmetric stretch of –CH_2_, which proved the existence of OA. In addition, the characteristic peaks belonging to the bare Fe_3_O_4_ surface at 3443 and 1630 cm^−1^ disappeared, and the new bands at 1531 and 1441 cm^−1^ appeared, corresponding to the symmetric and asymmetric stretch vibration of –COO^−^ of OA. According to Zhang [47], the type of interaction between the carboxylate and metal atom can be distinguished by calculating the wavenumber separation between these two peaks. As illustrated in Figure 4c, the wavenumber separation was 90 cm^−1^, less than 110 cm^−1^, indicating the existence of a bidentate bond. Additionally, the stretching vibration peak of C=O corresponding to free carboxylate appeared at 1710 cm^−1^, which indicated that a double-molecular layer on the surface of the sample was formed by the interaction of the hydrophobic tails owing to the excessive amounts of OA. The contact angle (Appendix A) proved that the prepared clusters had strong hydrophilicity in an alkaline environment, because the negatively charged –COO^−^ group of the outer physically adsorbed layer provided hydrophilic and electrostatic repulsion forces for the clusters. In the XPS spectrum (Appendix A), the three peaks with binding energies of 530.1, 531.2, and 532.6 eV can be fitted by the O 1 s. The strongest peak at 530.1 eV resulted from lattice oxygen in Fe_3_O_4_. The peaks at 532.6 and 531.2 eV correspond to the oxygen of the carboxyl in the physically adsorbed layer and the oxygen in the bidentate bond, respectively [48,49,50]. The obvious two steps of weight loss in the results of TGA (Figure 4b) indicated that the bilayer coating of OA on the surface of Fe_3_O_4_ clusters can be achieved by rapid-flow synthesis. Among them, a slight weight loss of approximately 0.1 wt% was found between 20 and 100 °C, which could be attributed to the evaporation of a part of the bound water. The weight loss between 100 and 325 °C (18.5 wt%) was caused by the degradation of OA physically adsorbed in the outer layer. The weight loss between 325 and 500 °C (17.7 wt%) could be attributed to the degradation of OA chemically adsorbed in the inner layer. Between 500 and 600 °C, the thermogravimetric curve was level, indicating that the surface-coated OA was degraded completely. (In contrast, XPS spectra and thermogravimetric analysis of bare Fe_3_O_4_ nanoparticles under the same conditions are supplemented in Appendix A, respectively).

From Figure 5, it can be seen that the coercivity and remanence of the samples were zero, exhibiting superparamagnetism with a saturation magnetization of approximately 60.5 emu/g, which was slightly lower than that of bare Fe_3_O_4_ nanoparticles (62.9 emu/g) under the corresponding conditions. This showed that the modification of OA would not change the superparamagnetic behavior of magnetic particles, but would reduce their saturation magnetization, due to the decrease of the effective atoms contributing to the magnetism of the samples. Considering the content of OA, the saturation magnetization of the magnetite core could reach 94.8 emu/g, which was even higher than the 92 emu/g of the bulk maghemite, indicating that the prepared clusters had high magnetic responsiveness.

In a word, Fe_3_O_4_ nanoclusters with a bilayer OA coating, narrow size distribution, and satisfactory saturation magnetization could be obtained within 30 s by performing continuous flow synthesis, in situ modification, and self-assembly.

### 3.2. Influence Factors on the Formation of Bilayer OA-Coated Fe_3_O_4_ Nanoclusters

To obtain an insight into the process of in situ modification and self-assembly of Fe_3_O_4_ nanoparticles into clusters, a series of experiments were carried out under various conditions. The experimental conditions and results are summarized in Table 1.

#### 3.2.1. Effect of the Residence Time

By changing the residence time 1 of the particles in the microchannel after co-precipitation, the evolution of the initial particle growing process could be explored. As shown in Figure 6a, the mean size of the clusters from unaged primary particles was 110.9 nm (PDI = 0.443), showing a three-peaks distribution with the existence of large aggregates. By increasing the residence time 1 to 6.28 s, the mean size of the clusters decreased to 71.1 nm, showing a narrow unimodal size distribution (PDI = 0.201), but there was still a small tailing. When the residence time 1 was prolonged to 18.84 s, the size of the clusters significantly decreased to 47.9 nm as the PDI value decreased to 0.165. As for the XRD patterns (Appendix A), with the increase of residence time 1 from 0 to 18.84 s, an obvious increase (45%) of the peak intensity could be found, proving the improvement of the crystallinity. The results showed that the primary particles formed by co-precipitation need to age for a certain time to reach a stable state. Otherwise, the unstable particles with more surface defects and covered with more –OH groups may result in a significant increase of the grafting of OA in the inner layer and accelerate the agglomeration rate, as well as be prone to forming unstable aggregates. In addition, prolonging the time of modification and self-assembly could produce clusters with smaller and more uniform sizes (Figure 6b). A possible explanation is that more OA was adsorbed on the outer surface of the cluster to form a denser double-layer coating in favor of cluster stabilization.

#### 3.2.2. Effect of the Amount of OA

As shown in Figure 7, when the concentration of OA was decreased from 0.1 to 0.060 M, the mean size of the clusters increased to 97.6 nm, and the PDI also increased to 0.183. When the concentration of OA was 0.060 M, a bimodal distribution was observed, including large clusters with a size of approximately 200 nm and small clusters with a size of approximately 50 nm. When the concentration of OA was further decreased to 0.035 M, the magnetic fluid was completely unstable, and the bulk aggregates precipitated out of the aqueous phase quickly. The results showed that there was a threshold value for the amount of OA. A small amount of OA might be insufficient to stabilize all clusters against aggregation, and an excessive amount of OA would cause unnecessary waste of raw materials and increase the cost of the post-treatment process. Based on the monolayer adsorption model proposed by Shen [51], when a single spherical Fe_3_O_4_ nanoparticle (supposing that all nanoparticles were 8.8 nm spheres) was completely coated with monolayer OA, the theoretical amount of OA in the OA-modified Fe_3_O_4_ nanoparticles was 22 wt%, while 44 wt% OA was required to form a dense double-layer OA coating. When the amount of OA was more than 22 wt%, monolayer adsorption on the surface of the particles was preferentially formed via chemical bonding, inducing the instability of the particles to form clusters. To reduce the surface energy in the aqueous phase, excessive OA was physically absorbed on the primary layer with the hydrophobic tails, and the outward hydrophilic group prevented the further aggregation of the clusters until a stable double-layer coating structure was formed. Since the binding rate could be accelerated by increasing the concentration of free OA in the aqueous phase, the coating speed of the outer layer OA could be controlled by adjusting the amount of OA to realize the effective control of the size of clusters.

#### 3.2.3. Effect of the pH

The isoelectric point of Fe_3_O_4_ was at pH = 6.5 [52]. Correspondingly, the pH of the aqueous environment might influence Fe_3_O_4_ nanoparticles in terms of electrostatic repulsion and surface properties and then affects the adsorption process of OA. We adjusted the concentration of ammonia hydroxide used in coprecipitation (the concentration of ammonia hydroxide needed for complete reaction with the iron source was 1.6 M) to investigate the effect of pH. It could be seen that the preparation process was very sensitive to the pH (Figure 8). With the decrease of pH from 10.3 to 9.7, the mean size of the cluster increased to 122.6 nm (PDI = 0.215) with a bimodal distribution. This could be attributed to the partial protonation of the hydrophilic –COO^−^ group of outer OA layer to the hydrophobic –COOH group, inducing further agglomeration of the clusters. Notably, it was found in experiments that, by bubbling with pure nitrogen or adding HCl solution until pH ≤ 7, the clusters could precipitate from the aqueous phase, during which the hydrophilic –COO^−^ group of the outer OA layer was completely transformed to the hydrophobic –COOH group without destruction of the bilayer-coated structure and the Fe_3_O_4_ core. The separated clusters could also be redispersed in n-hexane to realize the transfer of magnetic clusters from aqueous phase medium to oil phase medium, which might have potential applications in various fields.

#### 3.2.4. Effect of the Temperature

At room temperature, the as-prepared magnetic emulsion was not stable in the aqueous phase (Figure 9). The clusters agglomerated heavily and rapidly precipitated out from the solution during storage. When the temperature of the system increased to 35 °C, relatively stable clusters with a mean size of 103.2 nm were obtained, but the uniformity was poor. Even a bimodal distribution was found. When the temperature was increased to 60 °C, the diameter of the clusters decreased to 47.6 nm, which could remain stable for a long time. At this temperature, the physical adsorption process of the outer OA layer was greatly accelerated, and this could effectively control the aggregation process.

### 3.3. Proposed Mechanism of Bilayer OA-Coated Fe_3_O_4_ Nanocluster Formation

According to the above analysis and discussion, we attempted to propose a formation mechanism of bilayer OA-coated Fe_3_O_4_ nanoclusters with the combination of nano-precipitation, in situ modification, and self-assembly process, as shown in Figure 10. In an alkaline environment, the Fe_3_O_4_ nanoparticles prepared by co-precipitation could maintain stability in the aqueous phase for a short time, due to the strong electrostatic repulsion, providing a uniform assembly environment. Additionally, the terminal hydrophobic –COOH group of OA deprotonated to the hydrophilic –COO^−^ group, which was much more easily grafted to the hydroxyl group on the surface of the Fe_3_O_4_ nanoparticles. Thanks to the excellent mass transfer performance enabled by the micromixer, OA could be quickly adsorbed on the surface of the particles to form a monolayer coating, preventing the growth of the initial particles. Meanwhile, the action of the terminal hydrophobic chain of OA caused the instability and agglomeration of the modified particles to form clusters in the aqueous phase. Due to OA’s high surface energy in the aqueous medium, hydrophobic tails of –(CH_2_)_7_CH=CH(CH_2_)_7_CH_3_ of excessive OA rapidly oriented to the same hydrophobic tails of the modified clusters, forming an interdigitated bilayer structure. In addition, the negatively charged and hydrophilic –COO^−^ group of the outer OA layer was extended into the aqueous phase, restraining the uncontrollable growth of clusters and maintaining stable dispersion in the aqueous phase, due to the electrostatic repulsion and steric hindrance effect.

The mechanism of the bilayer OA-coated Fe_3_O_4_ nanoclusters formation implies that the in situ modification and self-assembly process are tandem processes that are greatly dependent on the difference between the adsorption rate of the double OA layers. Among them, the process of nanoparticle modification by the inner OA layer was a reaction-limited regime, which could be regarded as an instant process. The process of nanocluster stabilization by the outer OA layer was a diffusion-limited regime, and the adsorption rate was controlled by the amount of OA, the pH, and the temperature. Accordingly, due to the shortening of the distance between particles in the system with a high reactant concentration, the agglomeration caused by monolayer modification of OA was intensified, causing the particles agglomerate into large aggregates in a short time. Thanks to the intensified mixing performance of the micromixer, the second layer of OA could reach the surface of the clusters promptly to inhibit the uncontrollable growth of the cluster. In contrast, the mixing performance of the stirred tank was not ideal enough to control this process (Appendix A), owing to its poor mixing performance and uniformity of space–time. Furthermore, the product was prone to oxidation when exposed to an oxygen atmosphere for a long time in the batch process; additionally, introducing inert gas protection would bring extra energy consumption. Therefore, the highly efficient mixing and oxygen-free environment provided by micromixing technology created the conditions for the in situ modification and self-assembly of oxygen-sensitive nanoparticles, and this is more economical and executable for industrialization.

## 4. Conclusions

In this work, a facile and productive preparation method of Fe_3_O_4_ nanoclusters was provided by the combination of a membrane dispersion microreactor and in situ introduction of excessive OA. With the above continuous flow synthesis mode, the prepared Fe_3_O_4_ nanoclusters had the characteristics of narrow size distribution and relatively high saturation magnetization (up to 60.5 emu/g with 36.2% OA content). Based on the characterization of XRD, XPS, FTIR, TGA, and so on, the binding modes between the Fe_3_O_4_ nanoparticles and double OA layers were confirmed, indicating the effective interaction during a short period thanks to the enhanced mixing and confinement effect provided by the membrane dispersion microreactor. The effects of the residence time, amount of OA, pH, and temperature were studied comprehensively, and the mechanism of the in situ modification and self-assembly process was proposed. The method enabled the control over the adsorption rate of the double OA layers. As a result, flexible adjustment of the nanocluster size in the range of 50–100 nm was available. Compared with the multistep and hour-level preparation obtained in the batch process, the preparation of bilayer OA-coated Fe_3_O_4_ nanoclusters could be completed under a continuous and oxygen-free condition within 30 s. In a word, the method presented in our work brought a continuous synthesis of uniform Fe_3_O_4_ nanoclusters with the advantages of being effective, economic, controllable, operable, easy to scale up, and potentially suitable for industry. In addition, the synthesis approach could be readily generalized for efficient and continuous preparation of other nanoclusters.

## Figures and Tables

**Figure 1 nanomaterials-12-00350-f001:**
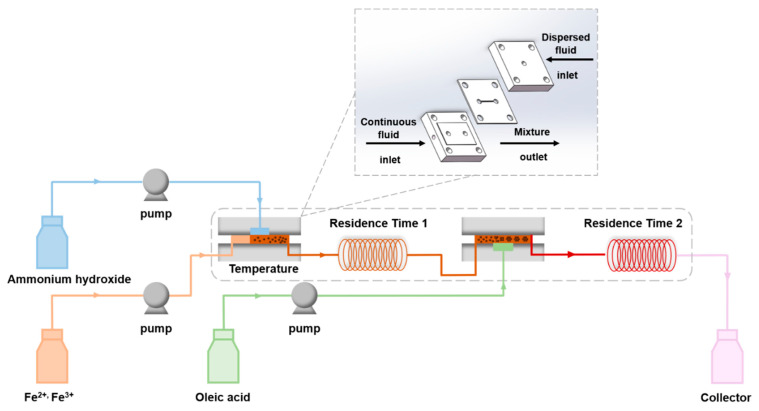
Schematic of the experimental setup.

**Figure 2 nanomaterials-12-00350-f002:**
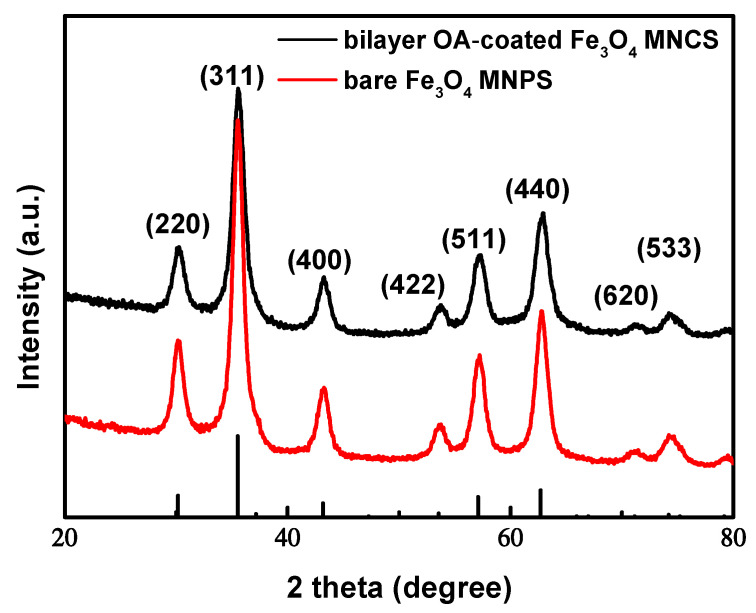
XRD pattern of bilayer OA-coated Fe_3_O_4_ nanoclusters and bare Fe_3_O_4_ nanoparticles under the same conditions.

**Figure 3 nanomaterials-12-00350-f003:**
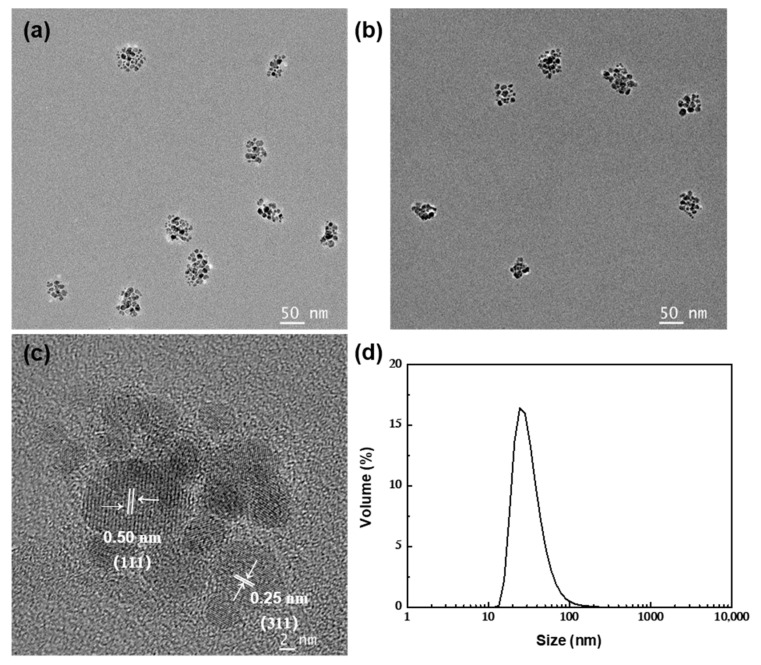
(**a**–**c**) TEM images of the bilayer OA-coated Fe_3_O_4_ nanoclusters and (**d**) size distribution.

**Figure 4 nanomaterials-12-00350-f004:**
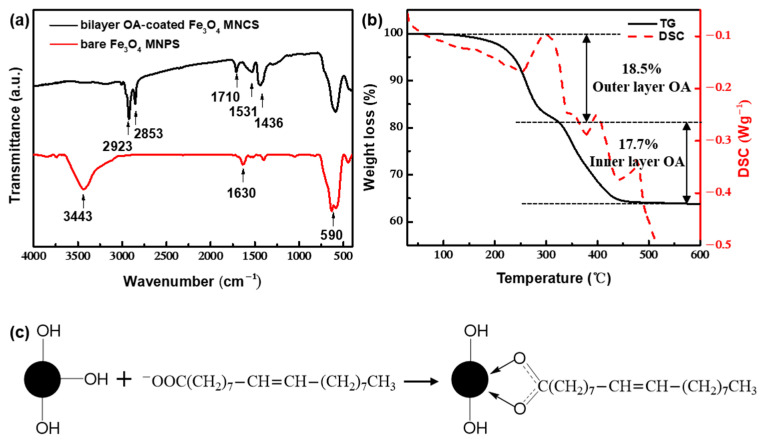
(**a**) FTIR spectra of bilayer OA-coated Fe_3_O_4_ nanoclusters and bare Fe_3_O_4_ nanoparticles under the same conditions. (**b**) Thermogravimetric analysis of bilayer OA-coated Fe_3_O_4_ nanoclusters. (**c**) Binding schemes for bidentate carboxylates.

**Figure 5 nanomaterials-12-00350-f005:**
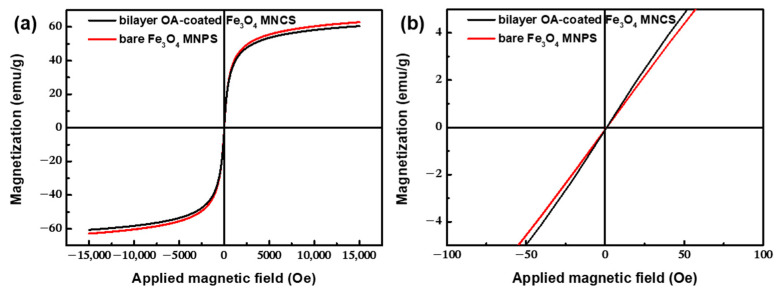
(**a**) Hysteresis loops of bilayer OA-coated Fe_3_O_4_ nanoclusters and bare Fe_3_O_4_ nanoparticles synthesized under the same conditions. (**b**) Magnification map of low magnetic field region.

**Figure 6 nanomaterials-12-00350-f006:**
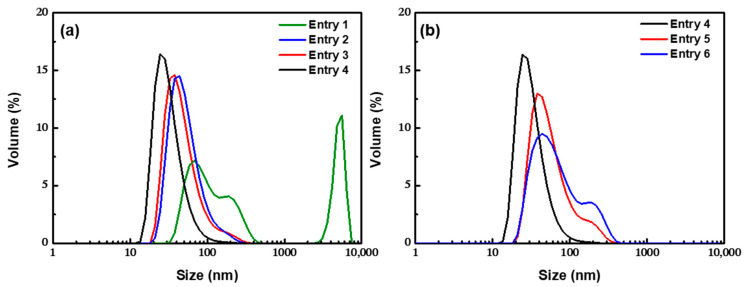
DLS profiles for nanoclusters obtained with (**a**) different residence time 1 and (**b**) residence time 2.

**Figure 7 nanomaterials-12-00350-f007:**
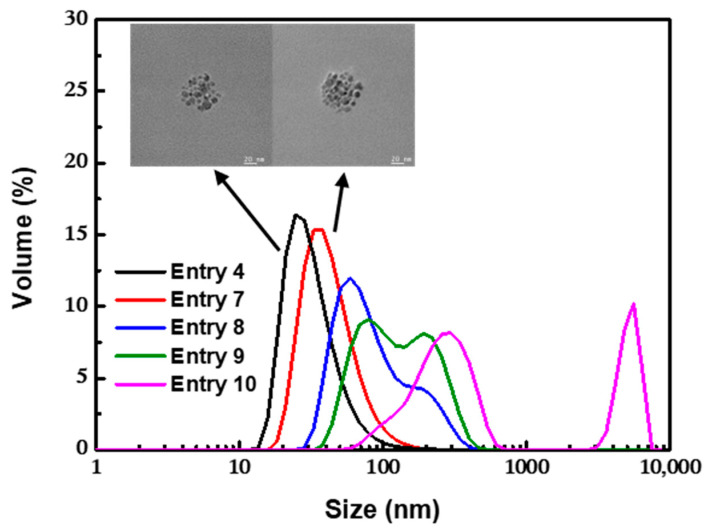
TEM images and DLS profiles for nanoclusters obtained with different amounts of OA. Scale bars indicate 20 nm.

**Figure 8 nanomaterials-12-00350-f008:**
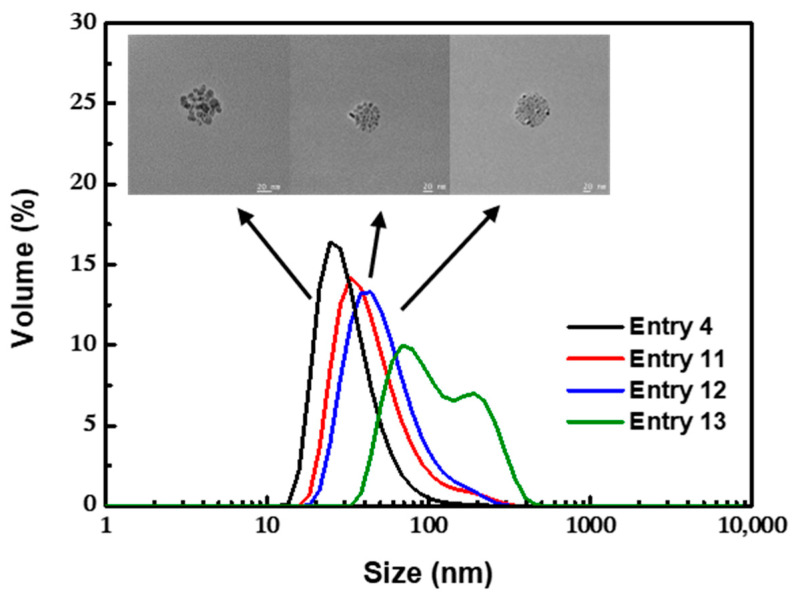
TEM images and DLS profiles for nanoclusters obtained with different pH values. Scale bars indicate 20 nm.

**Figure 9 nanomaterials-12-00350-f009:**
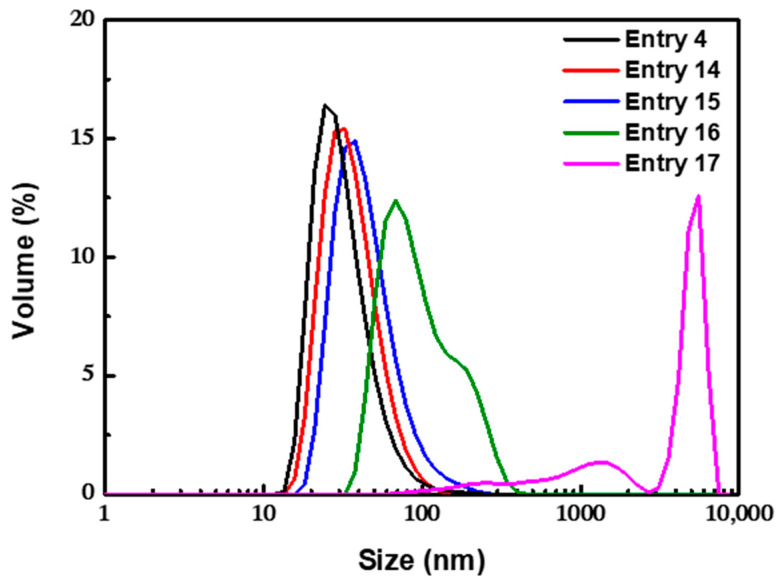
DLS profiles for nanoclusters obtained with different temperatures.

**Figure 10 nanomaterials-12-00350-f010:**
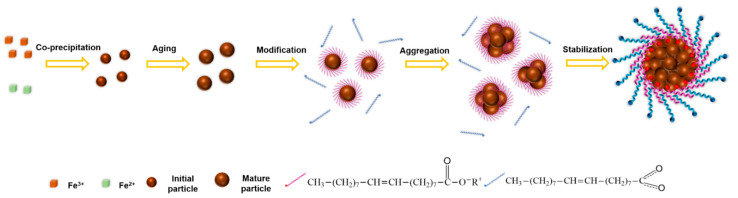
Schematic illustration of the formation processes of bilayer OA-coated Fe_3_O_4_ nanoclusters.

**Table 1 nanomaterials-12-00350-t001:** Summary of the operating conditions and sample characterization results for the continuous synthesis of bilayer OA-coated Fe_3_O_4_ nanoclusters.

Entry	NH_3_·H_2_O (M)	Oleic Acid (M)	Residence Time 1 (s)	Residence Time 2 (s)	pH	Temperature (°C)	Cluster Size (nm)	PDI
1	7.3	0.1	0	12.56	10.3	60	110.9	0.443
2	7.3	0.1	6.28	12.56	10.3	60	71.1	0.201
3	7.3	0.1	12.56	12.56	10.3	60	70.2	0.197
4	7.3	0.1	18.84	12.56	10.3	60	47.9	0.165
5	7.3	0.1	18.84	6.28	10.3	60	88.6	0.189
6	7.3	0.1	18.84	0	10.3	60	97.55	0.203
7	7.3	0.07	18.84	12.56	10.3	60	57.5	0.172
8	7.3	0.060	18.84	12.56	10.3	60	97.6	0.183
9	7.3	0.055	18.84	12.56	10.3	60	116.2	0.216
10	7.3	0.035	18.84	12.56	10.3	60	-	-
11	5.5	0.1	18.84	12.56	10.1	60	62.2	0.174
12	3.7	0.1	18.84	12.56	9.9	60	75.2	0.175
13	2.0	0.1	18.84	12.56	9.7	60	122.6	0.215
14	7.3	0.1	18.84	12.56	10.3	70	49.7	0.165
15	7.3	0.1	18.84	12.56	10.3	50	62.9	0.184
16	7.3	0.1	18.84	12.56	10.3	35	103.2	0.210
17	7.3	0.1	18.84	12.56	10.3	RT	-	-

## Data Availability

Not applicable.

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
