# Peer review of "Flexible and Effective Preparation of Magnetic Nanoclusters via One-Step Flow Synthesis"

_nanomaterials, 2022, doi:10.3390/nano12030350_

Round 1

Reviewer 1 Report

REFEREE REPORT

on paper Flexible and Effective Preparation of Magnetic Nanoclusters via One-Step Flow Synthesis

by authors Lin Zhou, Lu Ye and Yangcheng Lu

submitted to Nanomaterials

The paper Flexible and Effective Preparation of Magnetic Nanoclusters via One-Step Flow Synthesis is devoted to preparation and investigation of the Fe3O4 nanoclusters. A continuous co-precipitation, in-situ modification and self-assembly method were applied for the Fe3O4 nanoclusters fabrication. XRD, TEM, DLS, TGA, VSM, FTIR, XPS techniques were used for the samples investigation. The topic of this paper is critically actual. The work is of scientific interest to the audience of the Nanomaterials. The data are reliable and do not cause much doubt. Nevertheless, there are several points before the paper can be published. I hope that authors after major revisions can improve the paper and can publish it in Nanomaterials.

  1. I didn't find any information about direct application of such magnetic nanoclusters based on Fe3O4. My advice is to include more information about practical applications.
  2. The Introduction part must be improved with new relevant literature about magnetic nanomaterials and their applications in modern life. I suggest using relevant literature [please see and discuss: DOI: 10.3390/nano9040494; doi:10.4028/www.scientific.net/SSP.299.281; doi:10.4028/www.scientific.net/SSP.299.100].
  3. Why did you choose combination of continuous co-precipitation, in-situ modification and self-assembly methods for the samples preparation? What do them advantages in comparison with traditionally used?
  4. What was the aim to study the wetting properties of the clusters?
  5. Text in Fig. 1 is poorly read.
  6. What do the unmarked XRD peaks placed between 60 and 80 deg.?
  7. At which temperature the magnetization measurements of the nanoclusters were done? Did you evaluate the magnetic behavior between room and low temperatures?
  8. Insert in Fig. 7 doesn’t read; please improve the quality of this fragment. The same problem has the Fig. 13.
  9. The total number of Figures in the article is large; I think it would be better to make their number to a maximum of 10.
  10. Conclusion part is too short, please re-write it more widely.
  11. English must be improved, because now there are some typos and grammatical errors.

Reviewer 2 Report

Review of Nanomaterials-1563486, “Flexible and Effective Preparation of Magnetic Nanoclusters via One-Step Flow Synthesis”, by L. Zhou, L. Ye and Y. Lu.

My ability to review this manuscript is based on the fact that I have carried out similar studies. My ultimate view of this manuscript is that, while I am in basic agreement with the conclusions of the authors, I believe that an improved presentation is warranted. My comments are offered in that spirit.

Lines 29-30: Nanoparticles do not aggregate because of large surface-to-volume ratios, but because of surface-to-surface interactions. Indeed, the authors say so on lines 67-68. The fact that two different reasons were given suggests that two different authors wrote these sections, and no effort was made to correlate their contributions.

Section 3.1: There are several confusing issues in this section, particularly concerning the XPS and IR results. Concerning the XPS results, no figures are presented for the bare nanoparticles, so that we cannot see what changes occur when they are covered with oleic acid.

Further, while the fwhm values of components within a spectrum may vary under certain conditions, such conditions must be shown to apply here; further, when they do vary, they do not vary to the extent shown in the spectra. The present variations give the impression that the fwhm values were chosen to justify the authors’ conclusions.

Concerning the IR results, I believe that all references to “carboxylates” also apply to “carboxylic acids”. Further, the bidentate bond mentioned shown in Figure 5c (right hand side) gives a pentavalent carbon, which is impossible. Further, since the oleic acid is displaceable, I doubt that the bonding is covalent.

In addition, the TGA/DSC results would be more convincing if the data for the bare nanoparticles were also presented.

One final suggestion: since the TEM and DLS results are available, and the oleic acid length is known, can the calculated bilayer size estimate be compared with the DLS results?

Round 2

Reviewer 1 Report

Revised version can be accept as is